# Serum and Adipose Dipeptidyl Peptidase 4 in Cardiovascular Surgery Patients: Influence of Dipeptidyl Peptidase 4 Inhibitors

**DOI:** 10.3390/jcm11154333

**Published:** 2022-07-26

**Authors:** Ikuko Shibasaki, Toshiaki Nakajima, Taira Fukuda, Takaaki Hasegawa, Hironaga Ogawa, Go Tsuchiya, Yusuke Takei, Masahiro Tezuka, Takashi Kato, Yuta Kanazawa, Yasuyuki Kano, Toshiyuki Kuwata, Motoshi Ouchi, Shigeru Toyoda, Yoshimasa Aso, Hirotsugu Fukuda

**Affiliations:** 1Department of Cardiac and Vascular Surgery, Dokkyo Medical University School of Medicine, Mibu 321-0293, Tochigi, Japan; hironaga@dokkyomed.ac.jp (H.O.); gotsuchi@dokkyomed.ac.jp (G.T.); y-takei@dokkyomed.ac.jp (Y.T.); mtezuka@dokkyomed.ac.jp (M.T.); yk1223@dokkyomed.ac.jp (Y.K.); yasuyuki@dokkyomed.ac.jp (Y.K.); fukuda-h@dokkyomed.ac.jp (H.F.); 2Department of Medical KAATSU Training, Dokkyo Medical University School of Medicine, Mibu 321-0293, Tochigi, Japan; nakat@dokkyomed.ac.jp (T.N.); thasegawa6134@gmail.com (T.H.); 3Department of Cardiovascular Medicine, Dokkyo Medical University School of Medicine, Mibu 321-0293, Tochigi, Japan; s-toyoda@dokkyomed.ac.jp; 4Department of Liberal Arts and Human Development, Kanagawa University of Human Services, Yokosuka 238-8522, Kanagawa, Japan; fukuda-h9w@kuhs.ac.jp; 5Department of Cardiovascular Surgery, Maebashi Red Cross Hospital, Maebashi 371-0811, Gunma, Japan; takato@dokkyomed.ac.jp (T.K.); tkuwata@dokkyomed.ac.jp (T.K.); 6Department of Pharmacology and Toxicology, Dokkyo Medical University School of Medicine, Mibu 321-0293, Tochigi, Japan; ouchi@dokkyomed.ac.jp; 7Department of Endocrinology and Metabolism, Dokkyo Medical University School of Medicine, Mibu 321-0293, Tochigi, Japan; yaso@dokkyomed.ac.jp

**Keywords:** dipeptidyl peptidase 4, adiponectin, epicardial adipose tissue, subcutaneous adipose tissue

## Abstract

Dipeptidyl peptidase 4 (DPP-4) is a novel adipokine and may be involved in the association between adipose tissue and metabolic syndrome. We investigated DPP-4 and adiponectin levels in the serum, subcutaneous adipose tissue (SAT), and epicardial adipose tissue (EAT), and their relationship with preoperative factors, as well as comparing the DPP-4 levels in SAT and EAT with and without DPP-4 inhibitors. This study included 40 patients (25 men, age 67.5 ± 13.8 years). The serum adipokine, DPP-4, and adiponectin levels in SAT and EAT were measured using ELISA and Western blotting. The DPP-4 and adiponectin levels were significantly higher in the SAT than in the EAT. The serum DPP-4 and DPP-4 activity levels had no correlation with the DPP-4 levels in the SAT and EAT, but the DPP-4 levels in the SAT and EAT had a positive correlation. The DPP-4 levels in the SAT were positively correlated with atherosclerosis, diabetes mellitus, DPP-4-inhibitor use, and fasting blood glucose. The DPP-4 levels in the EAT showed a negative correlation with eGFR and a positive correlation with atrial fibrillation. The DPP-4 activity in the serum had a lower tendency in the group taking DPP-4 inhibitors than in the group not taking them. DPP-4 inhibitors may suppress angiogenesis and adipose-tissue hypertrophy.

## 1. Introduction

Currently, the number of patients with cardiovascular disease (CVD) remains high worldwide, including in Japan. CVD, primarily ischemic heart disease (IHD) and stroke [1], is the leading cause of death worldwide and is attributed to the adverse effects of cardiovascular risk factors, including aging and obesity [2]. Adipose tissue has been identified as a possible contributor to atherosclerosis and coronary artery disease (CAD) by promoting the development of endothelial and myocardial dysfunction and dyslipidemia [3,4,5]. Epicardial adipose tissue (EAT) is ectopic visceral fat around the heart that correlates with the incidence of conditions such as metabolic syndrome, coronary atherosclerosis, and atrial fibrillation (AF) and has been the subject of research on the biomarkers of cardiovascular risk [6].

Previously, we reported on adipokines affecting the EAT in patients with CAD [7], and various studies have reported that they have myocardium-protective and anti-inflammatory effects [8,9,10,11,12]. There have also been reports of the association of adipokines being with coronary artery calcification on computed tomography (CT) imaging [13,14,15], and of their utility as CVD and metabolic risk markers [16]. EAT is visceral fat deposited around the heart, especially around the epicardial coronary vessels. EAT protects the coronary arteries from the mechanical stresses of cardiac contraction and arterial pulse waves and acts as a support for coronary artery dilation [17]. By contrast, Wu et al. suggested that perivascular and EAT may have potentially detrimental effects on the coronary arteries, particularly the formation of atherosclerotic plaques [18].

Dipeptidyl peptidase 4 (DPP-4) is a 110-kilodalton cell-surface transmembrane protein, also known as CD26, which is expressed on the cell-plasma membranes of various tissues throughout the body [19,20]. Its expression is known to be particularly high in the kidneys [21]. The physiological function of DPP-4 is the rapid cleavage of the N-terminal dipeptides of incretin hormones (glucagon-like peptide-1 and glucose-dependent insu-linotropic polypeptide) and the subsequent inactivation of their insulinotropic activity, occurring within minutes [22,23]. Lamers et al. [24] identified DPP-4 as a novel adipokine released by fully differentiated human adipocytes. They reported that DPP-4 might be involved in the association between adipose tissue and metabolic syndrome. Sinitsky et al. [25] collected subcutaneous adipose tissue (SAT), EAT, and pericoronary adipose tissue. They reported a positive correlation between DPP-4 and very-low-density lipoprotein (VLDL) cholesterol concentrations in SAT.

Conversely, DPP-4 inhibitors for the treatment of diabetes mellitus (DM) were expected to have cardiovascular-protective effects, but large randomized clinical trials (EXAMINE [26], SAVOR-TIMI 53 [27]) reported only protection against cardiovascular events. In the CAROLINA study [28], DPP-4 inhibitors did not increase cardiovascular risk in Asian patients with type 2 DM. Similarly, a subgroup analysis of Asians showed that DPP-4 inhibitors have cardiovascular and renal safety profiles [29].

The purpose of this study was to investigate EAT, SAT, serum-adipokine-related DPP-4, adiponectin levels, and the effects of DPP-4 inhibitors on these parameters.

## 2. Materials and Methods

### 2.1. Study Population

Patients who underwent planned cardiovascular surgery with cardiopulmonary bypass (CPB) at the Dokkyo Medical University Hospital between October 2015 and June 2016 were included in this study. The study protocol was approved by the Dokkyo Medical University Hospital Ethics Committee (approval no: 27077) and was conducted in accordance with the Declaration of Helsinki. The need for written informed consent was waived owing to the retrospective nature of the study. Patients whose SAT or EAT could not be collected or could not be measured by Western blotting, even if it could be collected, were excluded.

Biochemical data were collected at the time of admission and analyzed using routine chemical methods in the clinical laboratory of the Dokkyo Medical University Hospital. Fasting blood glucose (FBG) and adipokines (adiponectin, leptin, DPP-4, and DPP-4 activity) were collected on the morning of the cardiovascular surgery. FBG was collected in tubes containing sodium ethylenediaminetetraacetic acid (EDTA) and polystyrene without anticoagulant. Serum was collected by centrifugation at 3000 rpm for 10 min at 4 °C and by centrifugation at 1000 rpm for 10 min at room temperature. To measure adipokine levels, blood samples were drawn into pyrogen-free tubes without EDTA. The serum was stored in aliquots at 80 °C for all enzyme-linked immunosorbent assays (ELISAs).

### 2.2. ELISA

Serum DPP-4 and adiponectin levels were measured using the Human DPP4/CD26 Assay Kit (27789, Immuno-Biological Laboratories Co., Ltd., Gunma, Japan) and Human Total Adiponectin/Acrp30 Quantikine ELISA Kit (DRP300, R&D Systems, Minneapolis, MN, USA). The serum DPP-4 activity was measured using a human DPP-4/CD26 Assay Kit for Biological Samples (Enzo Life Sciences Inc., Farmingdale, NY, USA). Serum leptin levels were also measured using the human Quantikine ELISA Kit (DLP00 for leptin, R&D Systems, Minneapolis, MN, USA).

### 2.3. Adipose Tissue Collection

Adipose tissue samples were obtained after initiation of CPB, as previously reported [6]. SAT samples were obtained from around the xiphoid process of the sternum, and EAT samples were collected near the proximal right coronary artery. Expression of DPP-4, adiponectin, and tumor necrosis factor α (TNFα) in subcutaneous and epicardial fat were assessed using Western blotting. The samples (100–150 mg) were homogenized in ice-cold lysis buffer (500-microliter protein elution buffer containing 8 µL protease inhibitor cocktail) and centrifuged at 14,000 rpm/25 min at 4 °C, and the supernatant was aliquoted. The specimens were stored at −80 °C. The protein concentration of each sample was measured using a Pierce BCA Protein Assay Kit (23227, Thermo Fisher Scientific Inc., Waltham, MA, USA). Next, SDS-polyacrylamide gel electrophoresis was performed using sample buffer (39001, Thermo Fisher Scientific Inc.) and polyvinylidene difluoride membrane (Amersham™ Hybond™ P PVDF, Cytiva, Marlborough, MA, USA), followed by blocking. Primary antibody reactions were performed overnight at 4 °C. The antibodies used in the assay were mouse anti-DPP4 monoclonal antibody (ab114033, Abcam plc, Cambridge, UK) for DPP-4, and mouse monoclonal anti-adiponectin (ab22554, Abcam). Secondary antibody reactions were performed at room temperature for 1 h. The antibodies used in the assay were DPP-4 and adiponectin goat anti-mouse IgG-HRP (sc-2005, Santa Cruz Biotech-nology, Inc., Dallas, TX, USA). Immunoreactive bands were visualized by chemiluminescence using Chemi-Lumi One Super (Chemi-Lumi One Super [02230-30, Nacalai Tesque, Kyoto, Japan]).

### 2.4. Statistical Analysis

Continuous variables are presented as means and standard deviations, and categorical variables are presented as counts and proportions. Spearman’s correlation coefficient was used to calculate the correlations. The differences in the characteristics between patient groups taking DPP-4 inhibitors and those not taking them were analyzed using the independent t-test (for normally distributed data) and Mann–Whitney U test (for non-normally distributed data, non-parametric test) for continuous variables, and χ^2^ test for categorical variables. Considering the small sample size, we analyzed the effect-size statistics γ (≥0.1: small, ≥0.3: medium, ≥0.5: large) using the Mann–Whitney U test. In the data (serum, SAT, EAT), outliers were measured using the inter-quartile range. Two or more outliers was excluded from the statistical analysis. All data were analyzed with SPSS software (version 22.0; IBM Corp., Armonk, NY, USA), and *p*-values < 0.05 were considered statistically significant.

## 3. Results

### 3.1. Patient Characteristics

The baseline characteristics of patients (*n* = 40) are shown in Table 1. This study included 25 men and 15 women. Their mean age was 67.5 ± 13.8 years, and body mass index (BMI) was 24.7 ± 4.4 kg/m^2^. The diagnoses were IHD in 13 patients (32.5%) and atherosclerosis, including IHD and aortic stenosis, in 17 patients (42.5%). The conventional risk factors, hypertension (HT), dyslipidemia, DM, and DPP-4 inhibitors use, were 29 (72.5%), 17 (42.5%), 15 (37.5%), and 10 (25%), respectively. The mean values for each were as follows: FBG, 117.8 ± 34.2 mg/dL; estimated glomerular filtration rate (eGFR), 62.3 ± 245.4 mL/min/1.73 m^2^; brain natriuretic peptide (BNP), 329.9 ± 490.7 pg/mL; hemoglobin A1c (HbA1c), 6.2 ± 0.9%; ejection fraction, 56.7 ± 9.0%; and coronary artery lesions, 2.3 ± 0.9. We further examined the characteristics of the patients with and without atherosclerotic disease (including IHD and aortic stenosis) (Appendix A).

### 3.2. Serum Adipokine Levels and Adipokine Protein Expression in Both SAT and EAT

The serum levels of the adipokines and the expression levels of the DPP-4 and the adiponectin in the adipocytes isolated from the EAT and SAT were measured (Table 1). Furthermore, when comparing the SAT and EAT, the DPP-4 and adiponectin were significantly higher in the SAT (*p* < 0.05 and *p* < 0.001, respectively), while the TNFα was higher in the EAT, although the difference was not significant (*p* = 0.069) (Figure 1).

### 3.3. Correlation between DPP-4 and Other Parameters in Serum, SAT, and EAT

Figure 2 shows the correlations of DPP-4 between serum, SAT, and EAT, and Table 2 shows the correlation between these and each parameter. First, the DPP-4 at the serum level showed a significant positive correlation (r = 0.618, *p* < 0.001) with the DPP-4 activity in the serum, but not with the DPP-4 levels in the SAT and EAT. The clinical data showed a significant negative correlation (r = −0.357, *p* < 0.05) with the HT. Second, the DPP-4 activity in the serum showed no correlation with the DPP-4 levels in the SAT and EAT, but a significant positive correlation (r = 0.419, *p* < 0.01, and r = 0.355, *p* < 0.05, respectively) with the adiponectin levels in the SAT and EAT. The clinical data, DM, and FBG levels showed a significant negative correlation (r = −0.333, *p* < 0.05; and r = −0.347, *p* < 0.05, respectively). Third, the DPP-4 levels in the SAT showed a significant positive correlation with the DPP-4 levels in the EAT (r = 0.327, *p* < 0.05), whereas they showed a significant negative correlation with the adiponectin levels in the SAT (r = 0.293, *p* < 0.05). The clinical data also showed a positive and significant correlation (r = 0.458, *p* < 0.01; r = 0.427, *p* < 0.05; r = 0.430, *p* < 0.01; r = 0.330, *p* < 0.05) between atherosclerosis, DM, DPP-4 inhibitors users, and FBG, respectively. Finally, the DPP-4 levels in the EAT showed a significant positive correlation with the adiponectin levels in the serum (r = 0.301, *p* < 0.05). The clinical data also showed a significant negative correlation with the eGFR (r = −0.319, *p* < 0.05) and a significant positive correlation with the AF (r = 0.323, *p* < 0.05).

### 3.4. Correlation between Adiponectin and Other Parameters in Serum, SAT, and EAT

Figure 3 shows the correlation of adiponectin between serum, SAT, and EAT. Table 3 shows the correlation between these and each parameter. First, the serum adiponectin levels showed a significant positive correlation with the adiponectin levels in the SAT and DPP-4 in EAT (r = 0.303, *p* < 0.05; r = 0.301, *p* < 0.05, respectively). Age, New York Heart Association (NYHA) score, AF, C-reactive protein, BNP level, and left atrial diameter were positively and significantly correlated (r = 0.268, *p* < 0.05; r = 0.435, *p* < 0.01; r = 0.4620 *p* < 0.01; r = 0.328, *p* < 0.05; r = 0.738 *p* < 0.01; r = 0.315, *p* < 0.05, respectively). By contrast, the insulin resistance (the ratio of triglycerides to HDL-cholesterol) and triglycerides showed a significant negative correlation (r = −0.428, *p* < 0.01; r = −0.460, *p* < 0.01, respectively). In addition, there was a positive correlation with hemodialysis creatinine and a negative correlation with eGFR, both of which are kidney-function indicators (r = 0.275, *p* < 0.05; r = 0.291, *p* < 0.05; r = −0.406, *p* < 0.01, respectively). Second, the adiponectin in the SAT showed a significant positive correlation with that in the EAT and the serum DPP-4 activity (r = 0.349, *p* < 0.05; r = 0.419, *p* < 0.01), whereas significant negative correlations with the DPP-4 were found in the SAT and serum leptin (r = −0.293, *p* < 0.05; r = −0.512, *p* < 0.01). The clinical data showed a positive correlation with NYHA (r = 0.347, *p* < 0.05) and significant negative correlations with aortic disease, DM, and DPP-4-inhibitor usage (r = −0.303, *p* < 0.05; r = −0.365, *p* < 0.05; r = −0.438, *p* < 0.01, respectively). Finally, the adiponectin in the EAT showed a significant negative correlation with the serum leptin (r = −0.615, *p* < 0.001) and a positive correlation with the serum DPP-4 activity and adiponectin in SAT (r = 0.355, *p* < 0.05; r = 0.341, *p* < 0.05, respectively). The clinical data showed negative and significant correlations with BMI, aortic disease, hypertension, total cholesterol, triglycerides, and aortic diameter (r = −0.289, *p* < 0.05; r = −0.439, *p* < 0.01; r = −0.279, *p* < 0.05; r = −0.274, *p* < 0.05; r = −0.292, *p* < 0.05; r = −0.328, *p* < 0.05, respectively). By contrast, a significant positive correlation was also found with NYHA (r = 0.297, *p* < 0.05).

### 3.5. Effects on Serum, SAT, and EAT between Patient Groups Taking DPP-4 Inhibitors and Those Not Taking Them

The comparison of the clinical characteristics and adipokine expression in the serum, SAT, and EAT between patient groups taking DPP-4 inhibitors and those not taking them is shown in Table 4. The group taking DPP-4 inhibitors had significantly higher DM, FBG, and HbA1c levels than that not taking DPP-4 inhibitors (all *p* < 0.01). There were no significant differences in the serum levels of adiponectin, leptin, and DPP-4 between the two groups. However, the DPP-4 activity in the serum had a lower tendency (*p* = 0.072), the DPP-4 level of SAT was significantly higher (*p* < 0.01, *γ* = 0.59), and the serum adiponectin level was significantly lower (*p* < 0.01, *γ* = 0.49) in the group taking DPP-4 inhibitors than those in the group not taking DPP-4 inhibitors. On the other hand, in the EAT, there was no significant difference between the two groups.

## 4. Discussion

The findings of the present study are as follows: (1) the DPP-4 and adiponectin levels were overexpressed in the SAT rather than in the EAT; (2) the DPP-4 and DPP-4 activity levels in the serum did not correlate with the DPP-4 levels in the SAT and EAT, but there was a positive correlation between the DPP-4 levels and the SAT and EAT; (3) the DPP-4 levels in the SAT showed a significantly positive correlation with atherosclerosis, DM, DPP-4-inhibitor use, and FBG levels. By contrast, the DPP-4 levels in the EAT showed a significant negative correlation with eGFR and a significant positive correlation with AF; and (4) in the group taking DPP-4 inhibitors, the DPP-4 levels in the SAT showed significantly higher values than in the group not taking DPP-4 inhibitors. However, the DPP-4 activity in the serum had a lower tendency in the group taking DPP-4 inhibitors than in those not taking them.

Our previous study [7] identified messenger ribonucleic acid (mRNA) levels of inflammatory cytokines, adipokines, neurohumoral factors, and their receptors in EAT, independent of the serum levels of these molecules, as well as the factors that influence the adipokines in EAT in patients with CAD.

DPP-4 was reported by Lamers D [19] et al. to be a novel adipokine that may decrease insulin sensitivity in an autocrine and paracrine fashion. DPP-4 is a ubiquitously expressed multifunctional type II membrane glycoprotein that is released from the serum membrane as a soluble peptide upon proteolysis [30,31]. These are established treatments for type-2 DM because they prevent the inactivation of GLP-1 [32]. The serum DPP-4 levels were positively correlated with adipocyte size and metabolic-syndrome indices, such as BMI, insulin, and leptin, and negatively correlated with age and adiponectin levels. Furthermore, DPP-4 expression is elevated in visceral fat in obesity, inflammatory conditions, and atherosclerosis [24]. In the present study, the DPP-4 and DPP-4 activity in the serum did not correlate with those in the SAT and EAT; this was similar to our previous study [7] in which the mRNA levels of inflammatory cytokines, adipokines, neurohumoral factors, and their receptors were unrelated to the serum levels of these molecules in EAT. The clinical data also showed that the serum DPP-4 levels were only associated with HT (r = −0.357; *p* < 0.05). These results suggest that serum DPP-4 levels may have a negative correlation with HT in cardiovascular-surgery patients, which is in accordance with the findings of a past study [33]. However, the serum DPP-4 activity was negatively associated with DM and FBG (r = −0.333; *p* < 0.05, r = −0.347; *p* < 0.05, respectively). Regarding this result, 37.5% of all the subjects had DM, and 67% were taking DPP-4 inhibitors. Additionally, the serum DPP-4 levels were positively correlated with the adiponectin levels in the SAT and EAT (r = 0.433, *p* < 0.01; r = 0.355, *p* < 0.05, respectively). By contrast, adiponectin produced by mature adipocytes is a good adipokine with insulin-sensitizing, antiatherogenic, antioxidant, and anti-inflammatory properties by stimulating the production of nitric oxide [34]. Low levels of serum adiponectin are associated with obesity, metabolic syndrome, DM, HT, and CAD [35], and high levels of serum adiponectin are associated with chronic heart failure, poor cardiac function, and high mortality, especially in patients with chronic kidney disease (CKD) [34,36]. These results were similar to those of the present study, in which serum adiponectin was positively correlated with NYHA, AF, and BNP, which are all implicated in heart failure. In addition, creatinine and hemodialysis were positively correlated and eGFR was negatively correlated, suggesting an association with renal failure. Furthermore, serum adiponectin, in contrast to DPP-4, was positively correlated with SAT and EAT adiponectin levels (r = 0.419; *p* < 0.01, r = 0.355; *p* < 0.05, respectively). The adiponectin levels in the SAT and EAT were positively correlated with serum adiponectin levels.

In the comparison of the expression levels of the SAT and EAT, the present study found that DPP-4 and adiponectin were significantly more expressed in the SAT, and the inflammatory cytokine TNFα, although not significantly, was more expressed in the EAT. Sinitsky et al. [25] reported that DPP-4 expression in EAT was three-fold higher than that in SAT, which is different from our results. Gruzdeva et al. [37] studied the expression of adipokines and cytokines in the adipocytes of EAT and SAT, dividing patients with CAD into two groups, according to the presence or absence of visceral obesity. Their results were similar to ours; the mRNA expression of adiponectin in the EAT of the patients with and without visceral obesity was lower than that in the SAT. By contrast, the expression of inflammatory cytokine genes (Interleukin-6 (IL6) and TNFα) was higher in the EAT than in the SAT; the EAT was associated with the presence of high levels of leptin and TNFα in the adipocytes and serum, as well as elevated lipid and carbohydrate metabolism. This is because when visceral adipose tissue (VAT) becomes dysfunctional, harmful adipose-acid species are deposited in additional adipose tissue, such as the myocardium, resulting in an increase in EAT [38]. When EAT expands, it becomes hypoxic and dysfunctional [39] and is invaded by macrophages and T lymphocytes, which increase the secretion of inflammatory cytokines, such as TNFα and IL6, contributing to the inflammatory environment characteristic of atherogenesis [40]. The expression of DPP-4 in SAT has been reported to be associated with serum VLDL cholesterol (r = 0.543; *p* < 0.05) [25]. In the present study, associations were demonstrated between DM, FBG, DPP-4-inhibitor use, and atherosclerosis.

The DPP-4 levels in the EAT were associated with those in the SAT, and the clinical data showed an association between eGFR and AF. Studies on EAT, atherosclerosis, and calcification in CKD are limited. EAT is associated with coronary-artery calcium and severity, a marker of atherosclerosis, and has also been reported to be associated with indicators of plaque instability detected on coronary CT in patients with CKD [41,42]. Among other measurements, the EAT levels in dialysis patients were shown to be higher than those in healthy subjects and increased in malnutrition, inflammation, and atherosclerosis syndromes [43]. A RIND trial subanalysis showed that EAT was an independent predictor of all-cause mortality after a median of 4 years [44]. These reports corroborate our results, showing a negative correlation (r = −0.319; *p* < 0.05) with eGFR. By contrast, obesity is a known risk factor for the development and progression of AF [45]. The extent and status of EAT have also been shown to be positively correlated with AF [45,46]. Krishnan et al. [47] reported that the adipokines secreted by adipocytes stimulate myofibroblast differentiation in association with the generation of reactive oxygen species, causing pronounced fibrosis in the EAT and myocardium. In addition, adipose tissue influences cardiomyocyte electrophysiology through adipokines. In our study, only DPP-4 in EAT was positively correlated with AF (r = 0.323; *p* < 0.05). DPP-4 inhibitors are taken for the treatment of type-2 DM to increase active GLP-1 levels and enhance and prolong the action of endogenously released incretin hormones [32].

Bilal et al. [48] reported the difference between SAT and VAT adipose tissue hypertrophy; the ability of the source cells of adipocytes (adipocyte progenitors (APs)) to proliferate and differentiate into adipocytes is high in SAT, leading to an increase in their number while remaining without hypertrophy and reducing the onset of DM. However, the APs in VAT were not only few in number, but also had low proliferative and differentiation potential, leading to an overload of existing adipocytes, hypertrophy, the induction of inflammation and insulin resistance, and the exacerbation of DM. Platelet-derived growth factor-B (PDGF-B) is the major factor that promotes angiogenesis in VAT by detaching pericytes from vessels, thereby causing vessel proliferation and the enlargement of the adipose tissue [42]. Watanabe et al. [49] reported that stromal-cell-derived factor 1 (SDF-1) inhibits PDGF-B-induced pericyte shedding. They found that SDF-1 in adipose tissue was increased in the obese state, but that both were degraded by an increase in DPP-4. Therefore, DPP-4 inhibitors suppressed the degradation of SDF-1 and inhibited pericyte shedding, angiogenesis, and adipose tissue hypertrophy. We further examined the expression of DPP-4 in the serum, SAT, and EAT in the groups taking DPP-4 inhibitors. The results showed no significant differences in DPP-4 levels in the serum and EAT, but DPP-4 activity in the serum had a lower tendency in the group taking DPP-4 inhibitors. Conversely, the DPP-4 levels in the SAT of the group taking DPP-4 inhibitors were significantly higher. Aso et al. [50] reported that DPP-4 enzyme activity was completely suppressed after 24 weeks of treatment with a DPP-4 inhibitor (teneligliptin). By contrast, serum DPP-4 antigen was significantly increased by DPP-4-inhibitor treatment compared with that of the baseline value. These reports suggest that DPP-4 inhibitors may inhibit adipose tissue hypertrophy in SAT.

This study had several limitations. First, the study included a small number of patients who underwent different types of cardiovascular surgery, and it had a low prevalence of patients taking DPP-4 inhibitors. Therefore, our findings may not necessarily be applicable to the general population of patients undergoing cardiovascular surgery. Secondly, hyperglycemia can be a confounding factor in the association of DPP-4-inhibitor usage with DPP-4 enzyme activity. Therefore, ideally, the FPG and HbA1c values of DPP-4-inhibitor users and non-users should be equalized when investigating DPP-4 enzyme activity. In addition, DPP-4 inhibitors were found to inhibit enzyme activity. However, in the present study, as for the SAT and EAT, the DPP-4 was measured by protein levels and not by enzyme activity. Therefore, further studies with a large number of patients and detailed analyses including DPP-4 enzyme activity levels are needed to clarify the effects of DPP-4 inhibitors on SAT and EAT.

## 5. Conclusions

This study showed that serum DPP-4 levels did not correlate with DPP-4 levels in SAT or EAT; the DPP-4 levels in the SAT correlated with the DPP-4 levels in the EAT, atherosclerosis, and FBG, and the DPP-4 levels in the EAT correlated with AF and eGFR. Further studies are necessary to determine the effects of DPP-4-inhibitor activity on SAT and EAT.

## Figures and Tables

**Figure 1 jcm-11-04333-f001:**
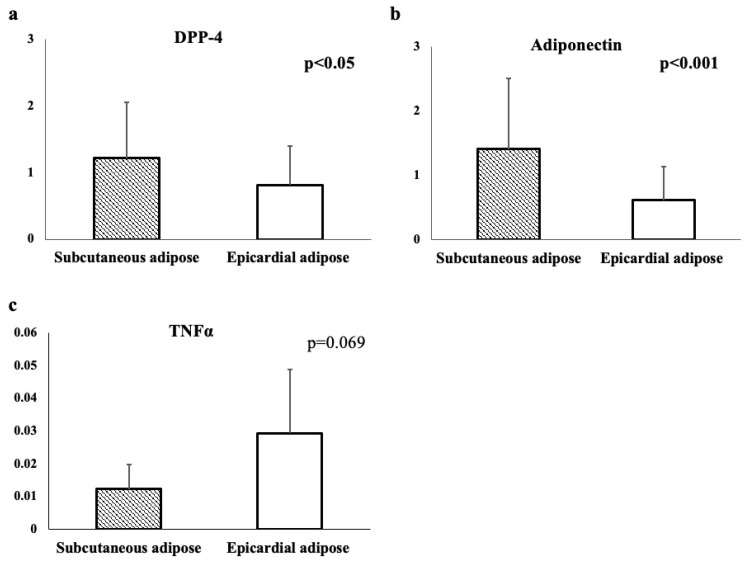
Comparison of dipeptidyl peptidase 4 (DPP-4), adiponectin, and tumor necrosis factor α (TNFα) expression in subcutaneous and epicardial adipose tissue. (**a**) Expression of DPP-4 in subcutaneous and epicardial adipose tissue. (**b**) Expression of adiponectin in subcutaneous and epicardial adipose tissue. (**c**) Expression of TNFα in subcutaneous and epicardial adipose tissue.

**Figure 2 jcm-11-04333-f002:**
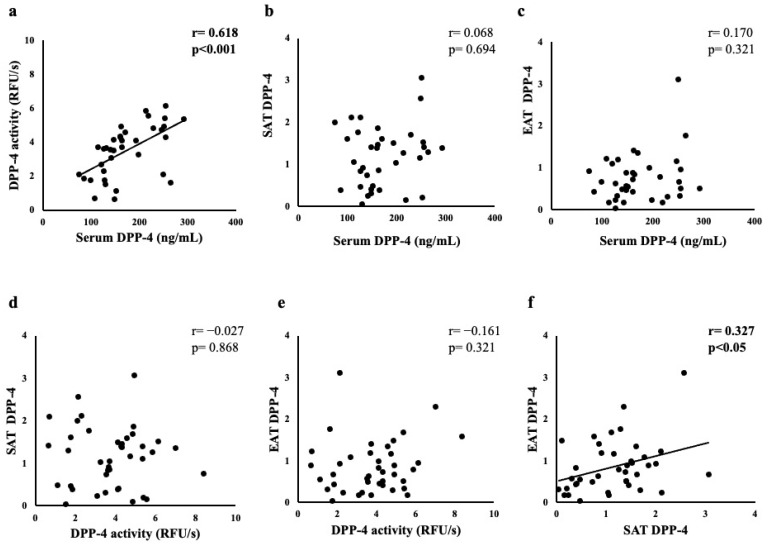
Correlations between (**a**) serum DPP-4 and serum DPP-4 activity, (**b**) serum and SAT DPP-4, (**c**) serum and EAT DPP-4, (**d**) serum DPP-4 activity and SAT DPP-4, (**e**) serum DPP-4 activity and EAT DPP-4, and (**f**) SAT and EAT DPP-4.

**Figure 3 jcm-11-04333-f003:**
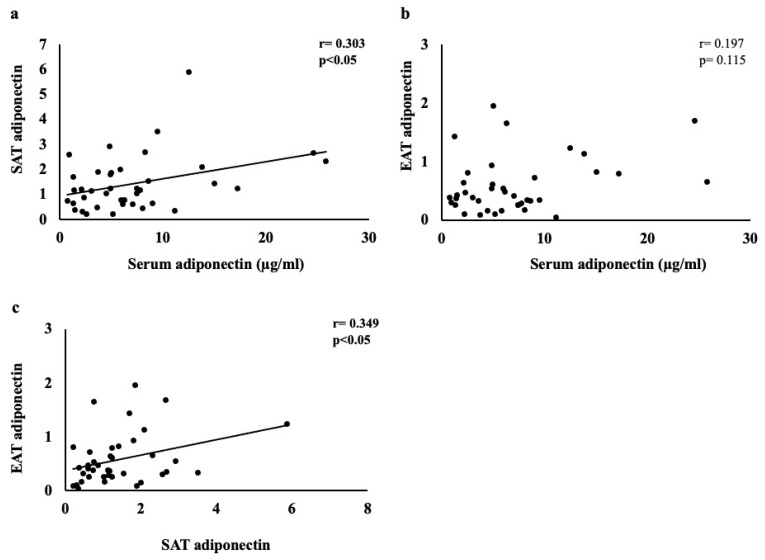
Correlations between (**a**) serum and SAT adiponectin, (**b**) serum and EAT adiponectin, and (**c**) SAT and EAT adiponectin.

**Table 1 jcm-11-04333-t001:** Patient characteristics.

Patients, Number	40
Male, *n* (%)	25 (62.5)
Age, years	67.5 ± 13.8
BMI, kg/m^2^	24.7 ± 4.4
Diagnosis	
Ischemic heart disease, *n* (%)	13 (32.5)
Valvular disease, *n* (%)	24 (60.0)
Aortic disease, *n* (%)	11 (27.5)
Atherosclerosis ^b^, *n* (%)	17 (42.5)
NYHA	2.1 ± 1.2
ECG	
Atrial fibrillation	11 (27.5)
Comorbidity	
Diabetes, *n* (%)	15 (37.5)
DPP-4 inhibitors (+), *n* (%)	10 (25.0)
Sitagliptin, *n* (%)	5 (50.0)
Teneligliptin, *n* (%)	3 (30.0)
Linagliptin, *n* (%)	1 (10.0)
Vildagliptin, *n* (%)	1 (10.0)
Hypertension, *n* (%)	29 (72.5)
Dyslipidemia, *n* (%)	17 (42.5)
Smoking, *n* (%)	5 (12.5)
Hemodialysis, *n* (%)	4 (10.0)
Fasting blood glucose, mg/dL	117.8 ± 34.2
Creatinine, mg/dL	1.25 ± 1.30
eGFR, mL/min/1.73 m^2^	62.3 ± 25.4
Total cholesterol ^a^, mg/dL	175.1 ± 40.2
High-density-lipoprotein cholesterol, mg/dL	51.8± 16.9
Low-density-lipoprotein cholesterol, mg/dL	98.2 ± 29.1
Triglycerides, mg/dL	114.5 ± 69.0
Triglycerides/high-density lipoprotein cholesterol	2.4 ± 1.6
C-reactive protein, mg/dL	1.0 ± 1.8
BNP, pg/mL	329.9 ± 490.7
Hemoglobin A1c, %	6.2 ± 0.9
TTE	
AoD, mm	32.5 ± 8.3
LAD, mm	44.2 ± 9.5
EF, %	56.7 ± 9.0
Number of coronary artery lesions, ischemic heart disease (13 patients)	2.3 ± 0.9
Serum levels	
Adiponectin, µg/mL	7.0 ± 5.9
Leptin ^a^, pg/mL	5712.6 ± 6095.4
DPP-4 ^a^, ng/mL	172.6 ± 57.6
DPP-4 activity, RFU/s	3.8 ± 1.7
Subcutaneous adipose	
DPP-4	1.15 ± 0.71
Adiponectin	1.41 ± 1.10
TNFα	2.32 ± 1.22
Epicardial adipose	
DPP-4	0.86 ± 0.63
Adiponectin	0.58 ± 0.47
TNFα	1.86 ± 1.15

The values shown are mean ± SD. ^a^ Missing values in total cholesterol (*n* = 1), leptin (*n* = 10), and DPP-4 (*n* = 4) were excluded. ^b^ Atherosclerosis includes ischemic heart disease and aortic valve stenosis. SD, standard deviation; BMI, body mass index; NYHA, New York Heart Association; ECG, electrocardiogram; BNP, brain natriuretic peptide; DPP-4, dipeptidyl peptidase 4; eGFR, estimated glomerular filtration; TTE, transesophageal echocardiography; AoD, aortic root diameter; LAD, left atrial dimension; EF, ejection fraction; RFU/s, relative fluorescent units per seconds; TNFα, tumor necrosis factor-α.

**Table 2 jcm-11-04333-t002:** Correlation between DPP-4 and other parameters in serum, subcutaneous adipose tissue, and epicardial adipose tissue.

	Spearman’s Rank
Serum DPP-4 level (ng/mL)	
BMI	0.109 (0.528)
Hypertension	−0.357 (0.033 *)
Triglycerides/high-density lipoprotein cholesterol	−0.068 (0.694)
Serum DPP-4 activity	0.618 (<0.001 ***)
Serum DPP-4 activity (RFU/s)	
BMI	−0.164 (0.312)
Diabetes mellitus	−0.333 (0.036 *)
Fasting blood glucose	−0.347 (0.028 *)
Triglycerides/high-density-lipoprotein cholesterol	−0.207 (0.199)
Serum DPP-4 level	0.618 (<0.001 ***)
Subcutaneous adipose adiponectin	0.433 (0.003 **)
Epicardial adipose adiponectin	0.355 (0.012 *)
Subcutaneous adipose tissue DPP-4	
BMI	0.180 (0.267)
Atherosclerosis ^a^	0.458 (0.003 **)
Diabetes mellitus	0.427 (0.016 *)
DPP-4 inhibitors	0.430 (0.006 **)
Fasting blood glucose	0.330 (0.044 *)
Triglycerides/high-density-lipoprotein cholesterol	−0.170 (0.296)
Epicardial adipose tissue DPP-4	0.327 (0.020 *)
Subcutaneous adipose tissue adiponectin	−0.293 (0.035 *)
Epicardial adipose tissue DPP-4	
BMI	0.169 (0.298)
Atrial fibrillation	0.323 (0.042 *)
Triglycerides/high-density lipoprotein cholesterol	−0.249 (0.121)
eGFR	−0.319 (0.045 *)
Subcutaneous adipose tissue DPP-4	0.327 (0.020 *)
Serum adiponectin level	0.301 (0.031 *)

Using Spearman’s rank correlation coefficient. * <0.05, ** <0.01, and *** <0.001. ^a^ Atherosclerosis includes ischemic heart disease and aortic valve stenosis. DPP-4, dipeptidyl peptidase 4; RFU/s, relative fluorescent units per seconds; BMI, body-mass index; eGFR, estimated glomerular filtration.

**Table 3 jcm-11-04333-t003:** Correlation between adiponectin and other parameters in serum, subcutaneous adipose tissue, and epicardial adipose tissue.

	Spearman’s Rank
Serum Adiponectin (µg/mL)	
Age	0.268 (0.049 *)
BMI	−0.276 (0.045 *)
Hemodialysis	0.291 (0.036 *)
NYHA	0.435 (0.003 **)
Atrial fibrillation	0.420 (0.004 **)
Creatinine	0.275 (0.045 *)
eGFR	−0.406 (0.005 **)
Triglycerides	−0.460 (0.002 **)
C-reactive protein	0.328 (0.021 *)
Triglycerides/high-density lipoprotein cholesterol	−0.428 (0.003 **)
BNP	0.738 (<0.01 **)
LAD	0.315 (0.027 *)
Subcutaneous adipose tissue adiponectin	0.303 (0.030 *)
Epicardial adipose tissue DPP-4	0.301 (0.031 *)
Subcutaneous adipose tissue adiponectin	
BMI	−0.215 (0.095)
Aortic disease	−0.303 (0.031 *)
NYHA	0.347 (0.015 *)
Diabetes mellitus	−0.365 (0.011 *)
DPP-4 inhibitors (+)	−0.438 (0.003 **)
Triglycerides/high-density-lipoprotein cholesterol	−0.033 (0.421)
Leptin	−0.512 (0.002 **)
DPP-4 activity	0.433 (0.003 **)
Subcutaneous-adipose-tissue DPP-4	−0.293 (0.035 *)
Epicardial adipose tissue adiponectin	0.341 (0.017 *)
Epicardial adipose tissue Adiponectin	
BMI	−0.289 (0.035 *)
Aortic disease	−0.439 (0.002 **)
NYHA	0.297 (0.031 *)
Hypertension	−0.279 (0.041 *)
Total cholesterol	−0.274 (0.046 *)
Triglycerides	−0.292 (0.034 *)
Triglycerides/high-density-lipoprotein cholesterol	−0.158 (0.165)
AoD	−0.328 (0.021 *)
Leptin	−0.615 (<0.001 ***)
DPP-4 activity	0.355 (0.012 *)
Subcutaneous adipose tissue Adiponectin	0.341 (0.017 *)

Using Spearman’s rank correlation coefficient. * <0.05, ** <0.01, and *** <0.001. BMI, body-mass index; NYHA, New York Heart Association; eGFR, estimated glomerular filtration; BNP, brain natriuretic peptide; LAD, left atrial dimension; DPP-4, dipeptidyl peptidase 4; AoD, aortic root diameter.

**Table 4 jcm-11-04333-t004:** Characteristics of patient groups taking DPP-4 inhibitors and those not taking them.

	DPP-4 Inhibitors (−)	DPP-4 Inhibitors (+)	*p*-Value ^c^	Effect Size ^e^
Patients, Number	30	10		
Male, *n* (%)	19 (63.3)	6 (60.0)	0.850	
Age, years	66.6 ± 15.2	70.0 ± 7.5	0.939 ^d^	
BMI, kg/m^2^	24.4 ± 4.6	25.5 ± 3.4	0.612 ^d^	
Diagnosis				
Ischemic heart disease, *n* (%)	11 (36.7)	6 (60.0)	0.196	
Valvular disease, *n* (%)	13 (43.3)	3 (30.0)	0.456	
Aortic disease, *n* (%)	6 (20.0)	1 (10.0)	0.471	
Atherosclerosis ^b^, *n* (%)	9 (30.0)	6 (60.0)	0.090	
NYHA	2.2 ± 1.2	1.8 ± 1.0	0.414 ^d^	
ECG				
Atrial fibrillation	9 (30.0)	2 (20.0)	0.540	
Comorbidity				
Diabetes, *n* (%)	5 (16.7)	10 (100)	<0.001 ***	
Hypertension, *n* (%)	20 (66.7)	9 (90.0)	0.152	
Dyslipidemia, *n* (%)	12 (40.0)	5 (50.0)	0.580	
Smoking, *n* (%)	3 (10.0)	2 (20.0)	0.408	
Hemodialysis, *n* (%)	4 (13.3)	0 (0)	0.224	
Fasting blood glucose, mg/dL	106.0 ± 20.4	152.9 ± 40.9	<0.001 ***^d^	
Creatinine, mg/dL	1.37 ±1.45	0.90 ± 0.20	0.747 ^d^	
eGFR, mL/min/1.73 m^2^	62.0 ± 27.2	63.0 ± 16.8	0.770 ^d^	
Total cholesterol ^a^, mg/dL	179.8 ± 36.5	161.6 ± 46.6	0.495 ^d^	
High-density-lipoprotein cholesterol, mg/dL	52.1 ± 13.2	50.9 ± 25.4	0.286 ^d^	
Low-density-lipoprotein cholesterol, mg/dL	102.2 ± 27.6	86.1 ± 30.2	0.148 ^d^	
Triglycerides, mg/dL	115.2 ± 64.7	120.5 ± 79.1	0.890 ^d^	
C-reactive protein, mg/dL	0.89 ± 1.77	1.14 ± 1.83	0.469 ^d^	
BNP, pg/mL	388.3 ± 538.0	154.7 ± 130.2	0.770 ^d^	
Hemoglobin A1c, %	5.9 ± 0.6	7.3 ± 0.8	<0.001 ***^d^	
TTE				
AoD, mm	32.7 ± 9.1	31.7 ± 3.1	0.961 ^d^	
LAD, mm	45.2 ± 10.3	41.1 ± 4.0	0.348 ^d^	
EF, %	56.7 ± 10.2	56.6 ± 8.7	0.842 ^d^	
Number of coronary-artery lesions, vessel disease	0.7 ± 1.2	1.1 ± 1.3	0.396^d^	
Serum levels				
Adiponectin, µg/mL	7.6 ± 6.4	5.3 ± 3.3	0.516	*γ* = 0.26
Leptin ^a^, pg/mL	6825.9 ± 7010.7	8543.3 ± 8702.9	0.368	*γ* = 0.10
DPP-4 ^a^, ng/mL	165.0 ± 50.2	192.5 ± 72.9	0.374	*γ* = 0.15
DPP-4 activity, RFU/s	4.1 ± 1.6	2.8 ± 1.9	0.072	*γ* = 0.29
Subcutaneous adipose				
DPP-4	0.98 ± 0.64	1.66 ± 0.68	0.006 **	*γ* = 0.59
Adiponectin	1.63 ± 1.17	0.74± 0.46	0.005 **	*γ* = 0.49
Epicardial adipose				
DPP-4	0.90 ± 0.68	0.73 ± 0.46	0.678	*γ* = 0.07
Adiponectin	0.58± 0.45	0.57 ± 0.55	0.740	*γ* = 0.02

The values are shown as mean ± SD. ** <0.01 and *** <0.001. ^a^ Missing values in total cholesterol (*n* = 1), leptin (*n* = 10), and DPP-4 (*n* = 4) were excluded. ^b^ Atherosclerosis includes ischemic heart disease and aortic valve stenosis. ^c^ Using Chi-square test or Fisher’s exact test. ^d^ Using Mann–Whitney U test. ^e^ Effect size was calculated by γ (≥0.1: small, ≥0.3: medium, ≥0.5: large) for Mann–Whitney. DPP-4, dipeptidyl peptidase 4; SD, standard deviation; BMI, body mass index; NYHA, New York Heart Association; ECG, electrocardiogram; BNP, brain natriuretic peptide; eGFR, estimated glomerular filtration; TTE, transesophageal echocardiography; AoD, aortic root diameter; LAD, left atrial dimension; EF, ejection fraction; RFU/s, relative fluorescent units per seconds.

## Data Availability

The data presented in this study are available on request from the corresponding author. The data are not publicly available due to part of future research.

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
