# Peer review of "Serum and Adipose Dipeptidyl Peptidase 4 in Cardiovascular Surgery Patients: Influence of Dipeptidyl Peptidase 4 Inhibitors"

_jcm, 2022, doi:10.3390/jcm11154333_

Round 1
Reviewer 1 Report
1.- About methods:
Clarify in the discussion, if there are data of deferential results in samples of SAT of abdominal origin (more related to metabolic syndrome and obesity)
2.- About study population:
The sample included a large number of patients with valvular diseases other than aortic stenosis (which can be considered equivalent to CV disease). Have the data from the two populations been analyzed separately?
Less than 25% of the subjects were being treated with DPP4 inhibitors, do you think this may affect the results? If the answer is whether to add to limitations.
The group that uses DPP4 inhibitors has worse fasting blood glucose and
glycosylated hemoglobin. Is it considered a confounding factor? Explain in discussion.
3.- About results:
In number of affected vessels has been taken from the total sample? An average of 0.8 does not make clinical sense (they do not need surgery), change by average of vessels in patients with coronary heart disease or atherosclerosis.
The total number of "diagnoses" is greater than 100 percent. Adjust and classify correctly in the table
Author Response
Reply to Reviewer 1
We greatly appreciate the attention you have given to our manuscript, and especially your excellent suggestions for improving the clarity and correctness of our intended message. We have corrected the paper as per your suggestions and consider that the revised manuscript has been greatly improved as a result.
Comments and Suggestions for Authors
1.- About methods:
Comment: Clarify in the discussion, if there are data of deferential results in samples of SAT of abdominal origin (more related to metabolic syndrome and obesity)
#) Answer: Thank you very much for your suggestion. Originally, we wanted to collect the subcutaneous abdominal fat and the fat from the large mesh of the stomach; however, this was not approved by our ethics committee. Therefore, we have taken subcutaneous fat and the fat around the xiphoid process of the sternum. We have rewritten this in more detail.
Page3, line121 “ SAT samples were obtained from around the xiphoid process of the sternum...”
2.- About the study population:
Comment: The sample included a large number of patients with valvular diseases other than aortic stenosis (which can be considered equivalent to CV disease). Have the data from the two populations been analyzed separately?
#) Answer: Thank you very much for your question. We have separately analyzed the data from the two populations. The results have been added to Supplementary table 1.
Page4, lines 165-167
“We further examined the characteristics of patients with and without atherosclerotic disease (including IHD and aortic stenosis) (Supplementary Table 1).”
Supplementary Table 1. Characteristics of patients with and without atherosclerotic disease
|
  |
Atherosclerotic diseasea(-) |
Atherosclerotic diseasea (+) |
p-valuec |
Effect sizee |
|
Patients, Number |
23 |
17 |
  |
  |
|
Male, n (%) |
16 (69.6) |
9 (52.9) |
0.283 |
  |
|
Age, years |
64.2 ± 15.2 |
71.9 ± 10.0 |
0.062d |
  |
|
BMI, kg/m2 |
24.8 ± 5.3 |
24.6 ± 2.6 |
0.914d |
  |
|
Diagnosis |
  |
  |
  |
  |
|
NYHA |
2.1 ± 1.2 |
2.1 ± 1.2 |
1.000d |
  |
|
ECG |
  |
  |
  |
  |
|
Atrial fibrillation |
10 (43.5) |
1 (5.9) |
0.008** |
  |
|
Comorbidity |
  |
  |
  |
  |
|
Diabetes, n (%) |
5 (21.7) |
10 (58.8) |
0.017* |
  |
|
DPP-4 inhibitors, n (%) |
4 (17.4) |
6 (35.3) |
0.196 |
|
|
Hypertension, n (%) |
16 (69.6) |
13 (76.5) |
0.629 |
  |
|
Dyslipidemia, n (%) |
6 (26.1) |
11 (64.7) |
0.015* |
  |
|
Smoking, n (%) |
3 (13.0) |
2 (11.8) |
0.904 |
  |
|
Hemodialysis, n (%) |
2 (8.7) |
2 (11.8) |
0.749 |
  |
|
Fasting blood glucose, mg/dL |
110.0 ± 26.7 |
128.2 ± 39.3 |
0.173d |
  |
|
Creatinine, mg/dL |
1.27 ±1.27 |
1.22 ± 1.31 |
0.448d |
  |
|
eGFR, mL/min/1.73 m2 |
61.3 ± 23.6 |
63.6 ± 27.1 |
0.665d |
  |
|
Total cholesterolb, mg/dL |
178.3 ± 37.0 |
172.3 ± 43.6 |
0.944d |
  |
|
High-Density Lipoprotein cholesterol, mg/dL |
52.7 ± 12.8 |
50.5 ± 21.2 |
0.277d |
  |
|
Low-Density Lipoprotein cholesterol, mg/dL |
98.3 ± 27.5 |
97.9 ± 31.3 |
0.914d |
  |
|
Triglycerides, mg/dL |
112.0 ± 71.9 |
117.9 ± 64.1 |
0.432d |
  |
|
Triglycerides/ High-Density Lipoprotein cholesterol |
2.2 ± 1.5 |
2.7 ± 1.7 |
0.277d |
  |
|
C-reactive protein, mg/dL |
1.17 ± 2.07 |
0.65 ± 1.21 |
0.371d |
|
|
BNP, pg/mL |
313.8 ± 411.6 |
351.7 ± 570.9 |
0.978d |
  |
|
Hemoglobin A1c, % |
6.01 ± 0.9 |
6.6 ± 0.9 |
0.024*d |
  |
|
TTE |
  |
  |
  |
  |
|
AoD, mm |
35.2 ± 9.0 |
28.6± 4.7 |
0.004**d |
  |
|
LAD, mm |
45.5 ± 9.5 |
42.5 ± 9.1 |
0.121d |
  |
|
EF, % |
55.8 ± 10.0 |
58.1± 9.5 |
0.665d |
  |
|
Number of coronary artery lesions, vessel disease |
0 |
1.8 ± 1.3 |
<0.001***d |
  |
|
Serum levels |
  |
  |
  |
  |
|
Adiponectin, µg/ml |
7.7 ± 6.6 |
11.5 ± 23.2 |
0.255d |
γ=0.15 |
|
Leptinb, pg/ml |
5440.2 ± 5557.5 |
6081.1 ± 6691.3 |
0.978d |
γ=0.06 |
|
DPP-4b, ng/ml |
166.1 ± 50.3 |
180.0 ± 65.7 |
0.851d |
γ=0.13 |
|
DPP-4 activity, RFU/sec |
3.8 ± 2.0 |
3.8 ± 1.2 |
0.808d |
γ=0.02 |
|
Subcutaneous adipose |
  |
  |
  |
  |
|
DPP-4 |
0.87 ± 0.58 |
1.52 ± 0.71 |
0.004**d |
γ=0.49 |
|
  Adiponectin |
1.55 ± 1.23 |
1.25± 0.87 |
0.289 |
γ=0.13 |
|
Epicardial adipose |
  |
  |
  |
  |
|
DPP-4 |
0.90 ± 0.62 |
0.80± 0.66 |
0.570d |
γ=0.09 |
|
  Adiponectin |
0.56 ± 0.49 |
0.68 ± 0.57 |
0.273d |
γ=0.12 |
The values are shown as mean ± SD. * <0.05, ** <0.01, and *** <0.001. aAtherosclerotic disease includes ischemic heart disease and aortic valve stenosis. bMissing values in total cholesterol (n=1), leptin (n=10), and DPP-4 (n=4) were excluded. cUsing Chi-square test or Fisher's exact test. dUsing Mann–Whitney U test. eEffect size was calculated by γ (≥0.1: small, ≥0.3: medium, ≥0.5: large) for Mann–Whitney. DPP-4, dipeptidyl peptidase 4; SD, standard deviation; BMI, body mass index; NYHA, New York Heart Association; ECG, electrocardiogram; BNP, brain natriuretic peptide; eGFR, estimated glomerular filtration; TTE, transesophageal echocardiography; AoD, aortic root diameter; LAD, left atrial dimension; EF, ejection fraction.
Comment: Less than 25% of the subjects were being treated with DPP4 inhibitors, do you think this may affect the results? If the answer is whether to add to limitations.
#) Answer:Thank you very much for your suggestion. The total number of patients was 40, which is a small sample size (15 DMs, 10 DPP-4 inhibitor users, and five non-users). Therefore, the present study is weak in terms of statistical analysis. However, since this is not a study targeting DM patients, I think it is unavoidable. Rather, it is expected that if the prevalence of DPP-4 inhibitor users increases, the DPP-4 enzyme activity would decrease, and the soluble DPP-4 level would increase in DPP-4 inhibitor users compared to non-users. We cannot deny that the low prevalence of DPP-4 inhibitor users in this study may affect the results, and we assume that increasing the percentage of DPP-4 inhibitor users would strengthen our results. We have added the following sentence to the Limitations
Page15, lines 382-384
“First, the study included a small number of patients who underwent different types of cardiovascular surgery, and had a low prevalence of patients taking DPP-4 inhibitors.”
Comment: The group that uses DPP4 inhibitors has worse fasting blood glucose and glycosylated hemoglobin. Is it considered a confounding factor? Explain in discussion.
#) Answer: Thank you very much for your comment. We have discussed these important points.
Until now, several studies have reported that chronic hyperglycemia and high HbA1c levels increase serum DPP-4 enzyme activity (1, 2). We have also reported that acute hyperglycemia increases soluble DPP-4 levels (3). Since all the users of DPP-4 inhibitors in this study were DM patients, it is unavoidable that both fasting plasma glucose level (FPG) and HbA1c are significantly higher than those of non-users.
References
- Plasma dipeptidyl peptidase-IV activity in patients with type-2 diabetes mellitus correlates positively with HbAlc levels, but is not acutely affected by food intake. Ryskjaer J, Deacon CF, Carr RD, Krarup T, Madsbad S, Holst J, Vilsbøll T.Eur J Endocrinol. 2006 Sep;155(3):485-93. doi: 10.1530/eje.1.02221.
- Hyperglycaemia increases dipeptidyl peptidase IV activity in diabetes mellitus. Mannucci E, Pala L, Ciani S, Bardini G, Pezzatini A, Sposato I, Cremasco F, Ognibene A, Rotella CM. Diabetologia. 2005 Jun;48(6):1168-72. doi: 10.1007/s00125-005-1749-8. Epub 2005 Apr 29.
- The serum level of soluble CD26/dipeptidyl peptidase 4 increases in response to acute hyperglycemia after an oral glucose load in healthy subjects: association with high-molecular weight adiponectin and hepatic enzymes. Aso Y, Terasawa T, Kato K, Jojima T, Suzuki K, Iijima T, Kawagoe Y, Mikami S, Kubota Y, Inukai T, Kasai K. Transl Res. 2013 Nov;162(5):309-16. doi: 10.1016/j.trsl.2013.07.011. Epub 2013 Aug 30.
Therefore, it cannot be denied that FPG and HbA1c are confounding factors. We have noted this in the Discussion.
Page15, lines 385-389
“Secondly, hyperglycemia can be a confounding factor in the association of DPP-4 inhibitor usage with DPP-4 enzyme activity. Therefore, ideally, the FPG and HbA1c values of DPP-4 inhibitor users and non-users should be equalized when investigating DPP-4 enzyme activity.”
3.- About results:
Comment: In number of affected vessels has been taken from the total sample? An average of 0.8 does not make clinical sense (they do not need surgery), change by average of vessels in patients with coronary heart disease or atherosclerosis.
#) Answer: Thank you very much for your comments. We have added the number of affected vessels. Page4, lines 164-165
“...and the mean number of coronary artery lesions in 13 patients with IHD was 2.3 ± 0.9.”
Page6, Table1
|
Patients, Number |
40 |
|
Male, n (%) |
25 (62.5) |
|
Age, years |
67.5 ± 13.8 |
|
BMI, kg/m2 |
24.7 ± 4.4 |
|
Diagnosis |
  |
|
Ischemic heart disease, n (%) |
13 (32.5) |
|
Valvular disease, n (%) |
24 (60.0) |
|
Aortic disease, n (%) |
11 (27.5) |
|
Atherosclerosisb, n (%) |
17 (42.5) |
|
NYHA |
2.1±1.2 |
|
ECG |
  |
|
Atrial fibrillation |
11 (27.5) |
|
Comorbidity |
  |
|
Diabetes, n (%) |
15 (37.5) |
|
DPP-4 inhibitors (+), n (%) |
10 (25.0) |
|
Sitagliptin, n (%) |
5 (50.0) |
|
Teneligliptin, n (%) |
3 (30.0) |
|
Linagliptin, n (%) |
1 (10.0) |
|
Vildagliptin, n (%) |
1 (10.0) |
|
Hypertension, n (%) |
29 (72.5) |
|
Dyslipidemia, n (%) |
17 (42.5) |
|
Smoking, n (%) |
5 (12.5) |
|
Hemodialysis, n (%) |
4 (10.0) |
|
Fasting blood glucose, mg/dL |
117.8 ± 34.2 |
|
Creatinine, mg/dL |
1.25 ± 1.30 |
|
eGFR, mL/min/1.73 m2 |
62.3 ± 25.4 |
|
Total cholesterola, mg/dL |
175.1 ± 40.2 |
|
High-Density Lipoprotein cholesterol, mg/dL |
51.8± 16.9 |
|
Low-Density Lipoprotein cholesterol, mg/dL |
98.2 ± 29.1 |
|
Triglycerides, mg/dL |
114.5 ± 69.0 |
|
Triglycerides/ High-Density Lipoprotein cholesterol |
2.4 ± 1.6 |
|
C-reactive protein, mg/dL |
1.0 ± 1.8 |
|
BNP, pg/mL |
329.9 ± 490.7 |
|
Hemoglobin A1c, % |
6.2 ± 0.9 |
|
TTE |
  |
|
AoD, mm |
32.5 ± 8.3 |
|
LAD, mm |
44.2 ± 9.5 |
|
EF, % |
56.7 ± 9.0 |
|
Number of coronary artery lesions, ischemic heart disease(13 patients) |
2.3 ± 0.9 |
|
Serum levels |
  |
|
Adiponectin, µg/ml |
7.0 ± 5.9 |
|
Leptina, pg/ml |
5712.6 ± 6095.4 |
|
DPP-4a, ng/ml |
172.6 ± 57.6 |
|
DPP-4 activity, RFU/sec |
3.8 ± 1.7 |
|
Subcutaneous adipose |
  |
|
DPP-4 |
1.15 ± 0.71 |
|
  Adiponectin |
1.41 ± 1.10 |
|
TNFα |
2.32 ± 1.22 |
|
Epicardial adipose |
  |
|
DPP-4 |
0.86 ± 0.63 |
|
  Adiponectin |
0.58 ± 0.47 |
|
TNFα |
1.86 ± 1.15 |
Comment: The total number of "diagnoses" is greater than 100 percent. Adjust and classify correctly in the table
#) Answer: Thank you very much for your suggestion. In recent years we have been performing combined surgeries, especially combining coronary artery bypass grafting with aortic valve replacement, and ascending aortic replacement with aortic valve replacement. Therefore, as you have pointed out, the percentage of diagnoses exceeds 100%.

Reviewer 2 Report
The article entitled “Serum and Adipose Dipeptidyl Peptidase 4 in Cardiovascular Surgery Patients: Influence of Dipeptidyl Peptidase 4 Inhibitors” highlighted the associations between DPP4 and adiponectin expression in subcutaneous and epicardial adipose tissues and demographic/laboratory parameters in individuals submitted to cardiovascular surgery. The manuscript is interesting and contributes to knowledge in the field. Before we proceed, please address the following points:
Major comments
1) In the Introduction section, briefly discuss the physiologic effects, site of production, and targets of DPP4.
2) Although the study was a retrospective analysis, written informed consent may be necessary. Were tissue and serum samples collected from a biobank from which written informed consent was previously obtained? Please clarify this matter.
3) As DPP4 inhibits GLP-1, which in turn, promotes more insulin secretion, less glucagon secretion and less gastric emptying, which ultimately leads to lower blood glucose, please provide information about the association between tissue/serum/activity levels of DPP4 and BMI and with a marker of insulin resistance, such as the ratio of triglycerides to HDL-cholesterol. Likewise, it would be interesting to have the analyses between serum/tissue levels of adiponectin and BMI and the ratio of triglycerides to HDL-cholesterol.
4) In addition, was the metabolic syndrome evaluated for the correlation of tissue/serum/activity levels of DPP4 and adiponectin?
5) In Figures 3A-C, were the analyses performed excluding the outlier? Were the same results found?
Minor comments
1) In Table 1, please correct de values of eGFR (it is written 62.3± 245.5)
2) Please describe which DPP4 inhibitor was used in the ten patients in the study.
Author Response
Reply to Reviewer 2
We greatly appreciate the attention you have given to our manuscript and especially your excellent suggestions for improving the clarity and correctness of our intended message. We have corrected the paper as per your suggestions and consider that the revised manuscript has been greatly improved.
The article entitled “Serum and Adipose Dipeptidyl Peptidase 4 in Cardiovascular Surgery Patients: Influence of Dipeptidyl Peptidase 4 Inhibitors” highlighted the associations between DPP4 and adiponectin expression in subcutaneous and epicardial adipose tissues and demographic/laboratory parameters in individuals submitted to cardiovascular surgery. The manuscript is interesting and contributes to knowledge in the field. Before we proceed, please address the following points:
Comments and Suggestions for Authors
Major comments
- In the Introduction section, briefly discuss the physiologic effects, site of production, and targets of DPP4.
#) Answer: Thank you very much for your suggestions. We have briefly discussed the above points. The manuscript has been revised as follows:
Page 2, lines 69-75
Dipeptidyl peptidase 4 (DPP-4) is a 110 kD cell surface transmembrane protein, also known as CD26, that is expressed on the cell plasma membrane of various tissues throughout the body [19,20]. The expression is known to be particularly high in the kidneys [21]. The physiological function of DPP-4 is the rapid cleavage of the N-terminal dipeptides of incretin hormones (glucagon-like peptide-1 and glucose-dependent insulinotropic polypeptide) and subsequent inactivation of their insulinotropic activity, occurring within minutes [22,23].
- Although the study was a retrospective analysis, written informed consent may be necessary. Were tissue and serum samples collected from a biobank from which written informed consent was previously obtained? Please clarify this matter.
#) Answer: We have addressed this issue in the Methods section.
Serum and adipose tissue samples were obtained during cardiac surgery, as shown in Table 1. The study protocol conformed to the Declaration of Helsinki and was approved by the institutional human research committee, and the proposal was approved by the Regional Ethics Committee of Dokkyo Medical University Hospital.
- As DPP4 inhibits GLP-1, which in turn, promotes more insulin secretion, less glucagon secretion and less gastric emptying, which ultimately leads to lower blood glucose, please provide information about the association between tissue/serum/activity levels of DPP4 and BMI and with a marker of insulin resistance, such as the ratio of triglycerides to HDL-cholesterol. Likewise, it would be interesting to have the analyses between serum/tissue levels of adiponectin and BMI and the ratio of triglycerides to HDL-cholesterol.
#) Answer: We have performed an analysis per your suggestion. Unfortunately, no significant differences were found for BMI and TG/HDL. This was added to Table 2.
|
  |
Spearman's rank |
|
Serum DPP-4 level (ng/ml) |
  |
|
BMI |
0.109 (0.528) |
|
Hypertension |
-0.357 (0.033*) |
|
Triglycerides /High-Density Lipoprotein cholesterol |
-0.068 (0.694) |
|
Serum DPP-4 activity |
0.618 (<0.001***) |
|
Serum DPP-4 activity (RFU/sec) |
  |
|
BMI |
-0.164 (0.312) |
|
Diabetes mellitus |
-0.333 (0.036*) |
|
Fasting blood glucose |
-0.347 (0.028*) |
|
Triglycerides /High-Density Lipoprotein cholesterol |
-0.207 (0.199) |
|
Serum DPP-4 level |
0.618 (<0.001***) |
|
Subcutaneous adipose Adiponectin |
0.433 (0.003**) |
|
Epicardial adipose Adiponectin |
0.355 (0.012*) |
|
Subcutaneous adipose tissue DPP-4 |
  |
|
BMI |
0.180 (0.267) |
|
Atherosclerosisa |
0.458 (0.003**) |
|
Diabetes mellitus |
0.427 (0.016*) |
|
DPP-4 inhibitors |
0.430 (0.006**) |
|
Fasting blood glucose |
0.330 (0.044*) |
|
Triglycerides /High-Density Lipoprotein cholesterol |
-0.170 (0.296) |
|
Epicardial adipose tissue DPP-4 |
0.327 (0.020*) |
|
Subcutaneous adipose tissue Adiponectin |
-0.293 (0.035*) |
|
Epicardial adipose tissue DPP-4 |
  |
|
BMI |
0.169 (0.298) |
|
Atrial fibrillation |
0.323 (0.042*) |
|
Triglycerides /High-Density Lipoprotein cholesterol |
-0.249 (0.121) |
|
eGFR |
-0.319 (0.045*) |
|
Subcutaneous adipose tissue DPP-4 |
0.327 (0.020*) |
|
Serum Adiponectin level |
0.301 (0.031*) |
- In addition, was the metabolic syndrome evaluated for the correlation of tissue/serum/activity levels of DPP4 and adiponectin?
#) Answer: We have performed the analysis per your suggestion. We found significant differences with respect to serum adiponectin and TG/HDL. We have modified Table 3 as follows:
|
  |
Spearman's rank |
|
Serum Adiponectin (µg/ml) |
  |
|
Age |
0.268 (0.049*) |
|
BMI |
-0.276 (0.045*) |
|
Hemodialysis |
0,291 (0.036*) |
|
NYHA |
0.435 (0.003**) |
|
Atrial fibrillation |
0.420 (0.004**) |
|
Creatinine |
0.275 (0.045*) |
|
eGFR |
-0.406 (0.005**) |
|
Triglycerides |
-0.460 (0.002**) |
|
C-reactive protein |
0.328 (0.021*) |
|
Triglycerides /High-Density Lipoprotein cholesterol |
-0.428 (0.003**) |
|
BNP |
0.738 (<0.01**) |
|
LAD |
0.315 (0.027*) |
|
Subcutaneous adipose tissue Adiponectin |
0.303 (0.030*) |
|
Epicardial adipose tissue DPP-4 |
0.301 (0.031*) |
|
Subcutaneous adipose tissue Adiponectin |
  |
|
BMI |
-0.215 (0.095) |
|
Aortic disease |
-0.303 (0.031*) |
|
NYHA |
0.347 (0.015*) |
|
Diabetes mellitus |
-0.365 (0.011*) |
|
DPP-4 inhibitors (+) |
-0.438 (0.003**) |
|
Triglycerides /High-Density Lipoprotein cholesterol |
-0.033 (0.421) |
|
Leptin |
-0.512 (0.002**) |
|
DPP-4 activity |
0.433 (0.003**) |
|
Subcutaneous adipose tissue DPP-4 |
-0.293 (0.035*) |
|
Epicardial adipose tissue Adiponectin |
0.341 (0.017*) |
|
Epicardial adipose tissue Adiponectin |
  |
|
BMI |
-0.289 (0.035*) |
|
Aortic disease |
-0.439 (0.002**) |
|
NYHA |
0.297 (0.031*) |
|
Hypertension |
-0..279 (0.041*) |
|
Total cholesterol |
-0.274 (0.046*) |
|
Triglycerides |
-0.292 (0.034*) |
|
Triglycerides /High-Density Lipoprotein cholesterol |
-0.158 (0.165) |
|
AoD |
-0.328 (0.021*) |
|
Leptin |
-0.615 (<0.001***) |
|
DPP-4 activity |
0.355 (0.012*) |
|
Subcutaneous adipose tissue Adiponectin |
0.341 (0.017*) |
Pages 4-5, lines 201-203
“In contrast, insulin resistance (ratio of triglycerides to HDL-cholesterol) and triglycerides showed a significant negative correlation (r=-0.428, p<0.01; r=-0.460, p<0.01, respectively).”
5) In Figures 3A-C, were the analyses performed excluding the outlier? Were the same results found?
#) Answer: Thank you very much for your questions. We did not analyze the excluded outliers. We have now performed the analysis again, excluding the abnormal values of serum adiponectin. In contrast to the previous results, there was no correlation between serum adiponectin levels and EAT adiponectin. Therefore, we have modified Figure 3 and Tables 1-3 in the revised manuscript.
Page12, Figure 3
Page 3, lines 149-151
“In the adiponectin data (serum, SAT, EAT), outliers were measured using the interquartile range. One case with two or more outliers was excluded from the statistical analysis.”
Page 4, lines 185-186
“, whereas they showed a significant negative correlation with adiponectin levels in SAT (r=0.293, p<0.05)”
Page 4, lines 189-190
“Finally, DPP-4 levels in EAT showed a significant positive correlation with adiponectin levels in the serum (r=0.301, p<0.05).”
Page 4, lines 196-198
“First, serum adiponectin levels showed a significant positive correlation with the adiponectin levels in SAT and DPP-4 in EAT (r=0.303, p<0.05; r=0.301, p<0.05, respectively).”
Page 5, lines 203-206
“In addition, there was a positive correlation with hemodialysis creatinine and a negative correlation with eGFR, both of which are kidney function indicators (r=0.275, p<0.05; r=0.291, p<0.05; r=-0.406, p<0.01, respectively).”
Page 5, lines 207-218
“...whereas significant negative correlations with DPP-4 were found in SAT and serum leptin (r=-0.293, p<0.05; r=-0.512, p<0.01). The clinical data showed a positive correlation with NYHA (r=0.347, p<0.05) and significant negative correlations with aortic disease, DM, and DPP-4 inhibitor usage (r=-0.303, p<0.05; r=-0.365, p<0.05; r=-0.438, p<0.01, respectively). Finally, adiponectin in EAT showed a significant negative correlation with serum leptin (r=-0.615, p<0.001) and positive correlation with serum DPP-4 activity and adiponectin in SAT (r=0.355, p<0.05; r=0.341, p<0.05, respectively). The clinical data showed negative and significant correlations with BMI, aortic disease, hypertension, total cholesterol, triglycerides, and aortic diameter (r=-0.289, p<0.05; r=-0.439, p<0.01; r=-0.279, p<0.05; r=-0.274, p<0.05; r=-0.292, p<0.05; r=-0.328, p<0.05, respectively). In contrast, a significant positive correlation was also found with NYHA (r=0.297, p<0.05).”
Page 5, lines 227-231
“ However, DPP-4 activity in the serum had a lower tendency (p=0.072), the DPP-4 level of SAT was significantly higher (p<0.01, γ=0.59), and serum adiponectin level was significantly lower (p<0.01, γ=0.49) in the group taking DPP-4 inhibitors than those in the group not taking DPP-4 inhibitors. On the other hand, in EAT, there was no significant difference between the two groups.”
Page 14, lines 314-316
“In addition, creatinine and hemodialysis were positively correlated and eGFR was negatively correlated, suggesting an association with renal failure. ”
Minor comments
- In Table 1, please correct de values of eGFR (it is written 62.3± 245.5)
#) Answer: Thank you very much. We have corrected it.
Page 5, Table1
eGFR, mL/min/1.73 m2 62.3 ± 25.4
2) Please describe which DPP4 inhibitor was used in the ten patients in the study.
#) Answer: We have described it, please see Table 1 below.
|
Patients, Number |
40 |
|
Male, n (%) |
25 (62.5) |
|
Age, years |
67.5 ± 13.8 |
|
BMI, kg/m2 |
24.7 ± 4.4 |
|
Diagnosis |
  |
|
Ischemic heart disease, n (%) |
13 (32.5) |
|
Valvular disease, n (%) |
24 (60.0) |
|
Aortic disease, n (%) |
11 (27.5) |
|
Atherosclerosisb, n (%) |
17 (42.5) |
|
NYHA |
2.1±1.2 |
|
ECG |
  |
|
Atrial fibrillation |
11 (27.5) |
|
Comorbidity |
  |
|
Diabetes, n (%) |
15 (37.5) |
|
DPP-4 inhibitors (+), n (%) |
10 (25.0) |
|
Sitagliptin, n (%) |
5 (50.0) |
|
Teneligliptin, n (%) |
3 (30.0) |
|
Linagliptin, n (%) |
1 (10.0) |
|
Vildagliptin, n (%) |
1 (10.0) |
|
Hypertension, n (%) |
29 (72.5) |
|
Dyslipidemia, n (%) |
17 (42.5) |
|
Smoking, n (%) |
5 (12.5) |
|
Hemodialysis, n (%) |
4 (10.0) |
|
Fasting blood glucose, mg/dL |
117.8 ± 34.2 |
|
Creatinine, mg/dL |
1.25 ± 1.30 |
|
eGFR, mL/min/1.73 m2 |
62.3 ± 25.4 |
|
Total cholesterola, mg/dL |
175.1 ± 40.2 |
|
High-Density Lipoprotein cholesterol, mg/dL |
51.8± 16.9 |
|
Low-Density Lipoprotein cholesterol, mg/dL |
98.2 ± 29.1 |
|
Triglycerides, mg/dL |
114.5 ± 69.0 |
|
Triglycerides/ High-Density Lipoprotein cholesterol |
2.4 ± 1.6 |
|
C-reactive protein, mg/dL |
1.0 ± 1.8 |
|
BNP, pg/mL |
329.9 ± 490.7 |
|
Hemoglobin A1c, % |
6.2 ± 0.9 |
|
TTE |
  |
|
AoD, mm |
32.5 ± 8.3 |
|
LAD, mm |
44.2 ± 9.5 |
|
EF, % |
56.7 ± 9.0 |
|
Number of coronary artery lesions, ischemic heart disease(13 patients) |
2.3 ± 0.9 |
|
Serum levels |
  |
|
Adiponectin, µg/ml |
7.0 ± 5.9 |
|
Leptina, pg/ml |
5712.6 ± 6095.4 |
|
DPP-4a, ng/ml |
172.6 ± 57.6 |
|
DPP-4 activity, RFU/sec |
3.8 ± 1.7 |
|
Subcutaneous adipose |
  |
|
DPP-4 |
1.15 ± 0.71 |
|
  Adiponectin |
1.41 ± 1.10 |
|
TNFα |
2.32 ± 1.22 |
|
Epicardial adipose |
  |
|
DPP-4 |
0.86 ± 0.63 |
|
  Adiponectin |
0.58 ± 0.47 |
|
TNFα |
1.86 ± 1.15 |

Round 2
Reviewer 2 Report
The authors addressed all comments. Congratulations on your work.